# Mantle-Derived Noble Gas Isotopes in the Ore-Forming Fluid of Xingluokeng W-Mo Deposit, Fujian Province

**Yun Gao** [1,2] , **Bailin Chen** [2,*], **Liyan Wu** [3,*], **Jianfeng Gao** [3] , **Guangqian Zeng** [2] **and Jinghui Shen** [2]

[1] School of Earth Sciences, China University of Geosciences, Wuhan 430074, China; gaoyun_10@163.com
[2] Key Laboratory of Paleomagnetism and Tectonic Reconstruction, Institute of Geomechanics, Chinese Academy of Geological Sciences, Beijing 100081, China; zengguangqian90@163.com (G.Z.); wxkssjh@email.cugb.edu.cn (J.S.)
[3] State Key Laboratory of Ore Deposit Geochemistry, Institute of Geochemistry, Chinese Academy of Sciences, Guiyang 550081, China; gaojianfeng@mail.gyig.ac.cn
[*] Correspondence: cblh6299@263.net (B.C.); wuliyan@mail.gyig.ac.cn (L.W.)

**Abstract:** China has the largest W reserves in the world, which are mainly concentrated in south China. Although previous studies have been carried out on whether mantle material is incorporated in granites associated with W deposits, the conclusions have been inconsistent. However, rare gas isotopes can be used to study the contribution of mantle-to-W mineralization. In this paper, we investigated the He and Ar isotope compositions of fluid inclusions in pyrite and wolframite from the Xingluokeng ultra-large W-Mo deposit to evaluate the origin of ore-forming fluids and discuss the contribution of the mantle-to-tungsten mineralization. The He-Ar isotopic compositions showed that the $^3He/^4He$ ratios of the ore-forming fluid of the Xingluokeng deposit ranged from 0.14 to 1.01 Ra (Ra is the $^3He/^4He$ ratio of air, 1 Ra = $1.39 \times 10^{-6}$), with an average of 0.58 Ra, which is between the $^3He/^4He$ ratios of mantle fluids and crustal fluids, suggesting that the mantle-derived He was added to the mineralizing fluid, with a mean of 8.7%. The $^{40}Ar/^{36}Ar$ ratios of these samples ranged from 361 to 817, with an average of 578, between the atmospheric $^{40}Ar/^{36}Ar$ and the crustal and/or mantle $^{40}Ar/^{36}Ar$. The results of the He-Ar isotopes from Xingluokeng W-Mo deposit showed that the ore-forming fluid of the deposit was not the product of the evolution of pure crustal melt. The upwelling mantle plays an important role in the formation of tungsten deposits.

**Keywords:** He and Ar isotopes; ore-forming fluids; mantle upwelling; Xingluokeng W-Mo deposit; south China



## 1. Introduction

China has the largest tungsten reserves in the world, with approximately 10.3 million tons of $WO_3$ [1,2], and most of the W deposits are mainly clustered in south China (Figure 1), which accounts for about 80% of the country's total and around 50% of the world's total [3–8]. The formation of W deposits in south China is mainly related to the Mesozoic (concentrated in 160~150 Ma) magmatic activity [9,10], and most of the W deposits in south China are concentrated in the Nanling metallogenic belt (NLMB). Traditionally, most W-related granites are considered as S-type granites, derived from the reworking of ancient meta-sedimentary rocks [11], and it is thus assumed that tungsten mineralization occurred without the addition of mantle material, but recent studies suggest that these granites may be mainly I-type granites [12,13]. The reason for the controversy is that the granites associated with tungsten mineralization are mainly high-differentiated granites, which makes it difficult to classify the genetic types and evaluate the influence of mantle on tungsten mineralization.

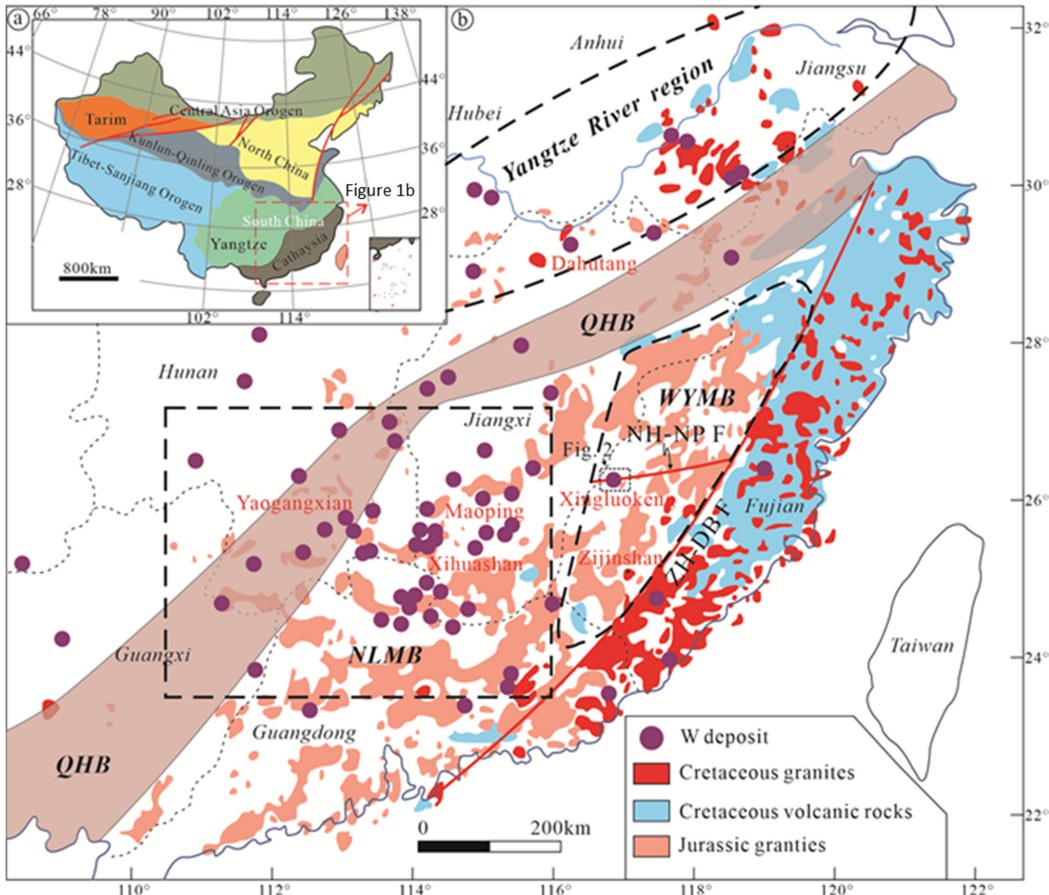

**Figure 1.** (**a**) Tectonic map of China. (**b**) Geologic map showing the distribution of Mesozoic granitoids and major tungsten deposits in south China (modified after [14–16]). NLMB = Nanling metallogenic belt, QHB = Qin-Hang metallogenic belt, WYB = Wuyi metallogenic belt, ZH-DB F = Zhenghe-Dabu fault, NH-NP F = Ninghua-Nanping fault.

Due to the distinct $^3$He/$^4$He ratios of the crust (~0.05–0.01 Ra, where Ra is the atmospheric $^3$He/$^4$He ratio, $1.39 \times 10^{-6}$) and upper mantle (6–9 Ra), and different $^{40}$Ar/$^{36}$Ar ratio of the atmosphere (295.5) versus the crust or mantle, the He-Ar isotopic compositions of fluid inclusion in minerals have been widely used in the studies of contribution of mantle-derived components and the relationship between crust–mantle interactions and mineralization [17–21]. Recently, a growing number of studies have used the method of He-Ar isotopes to prove that mantle-derived components were involved in the genesis of some W deposits (such as the Gerês W deposit [22] and Panasqueira W deposit [20] in Portugal, Dae Hwa W deposit in South Korea [23], and Xihuashan [24], Yaogangxian [18], Shizhuyuan [25], Piaotang [26], Taoxikeng [27], Yaoling-Meiziwo [28], and other W deposits in south China).

The Xingluokeng W-Mo polymetallic deposit is one of the superlarge W deposits in China and the largest W deposit in the Wuyi metallogenic belt (WYMB), which is historically a Cu-dominated polymetallic belt [10,29]. Many researchers have studied the petrographic geochemical characteristics [30–32], deposit geochemical characteristics [33,34], and geochronology [2,35]. However, whether mantle components are involved in the genesis of the deposit is poorly studied understood. In this paper, we investigated He-Ar isotopes of the ore fluids trapped in pyrite and wolframite from the Xingluokeng W-Mo polymetallic deposit, in order to reveal the genesis of the deposit and contribution of mantle-derived volatiles, providing useful information for understanding the granite-related tungsten metallogeny in the WYMB.

## 2. Geological Background

The south China block is composed of the Yangtze block to the northwest and the Cathaysia block to the southeast, separated by the Qin-Hang belt [36]. Around 900 Ma ago, the paleo South China Ocean gradually closed, resulting in the collision of Yangtze and Cathaysian blocks on both sides of the ocean basin, forming an arc-shaped orogenic belt with a width of more than 100 km and an extension of about 1500 km at the junction, i.e., Qin-Hang belt [37,38]. The geological background of south China is complex, with strong tectonic movements [39,40] and a wide distribution of granites of different ages and types, which are known for large-scale magmatic activities and mineralization events in the Mesozoic, forming huge deposits of W, Sn, U, and REE [7,41–45].

The WYMB is located in the northeast of the Cathaysian block and distributed in the NNE direction (Figure 1), bounded on the north by the Shaoxing-Jiangshan-Pingxiang fault and adjacent to the Qinhang metallogenic belt on the southeast edge of the Yangtze block, on the west by the Yingtan–Anyuan fault, near the NLMB, and on the east by the Lishui-Zhenghe-Dapu fault to the southeast coastal metallogenic belt [46,47]. The WYMB has experienced major geological events such as the formation and cracking of the Cathaysian block, the collision and splicing of the Cathaysian block and the Yangtze block, the collision between the north China block and south China block, and the subduction of the Pacific plate to the Eurasian continental margin, among which the Yanshanian tectonic magmatic activity is the strongest [48,49], and forming a large number of Cu, Au, Ag, Pb, and Zn deposits [50–53], such as the Zijinshan Cu–Au deposit and the Lengshuikeng Ag–Pb–Zn deposit, as well as some large W deposits, such as the Shangfang W deposit and Xingluokeng W-Mo deposit. The strata in WYMB include the upper Archean to Quaternary, in which the pre-Devonian is the basement rock series, the Devonian Middle Triassic is the caprock rock series dominated by marine sedimentation, and the Meso Cenozoic is the continental clastic and volcanic rock series. The fault is mainly NNE-trending, followed by NW-trending, and partly NEE-trending. The study area is located in the western part of the NEE-trending Nanping-Ninghua fault zone (Figure 1). In addition to the Xingluokeng deposit, there are also tungsten deposits and mineralized sites in the zone, such as Beikeng and Guomuyang (Figure 2).

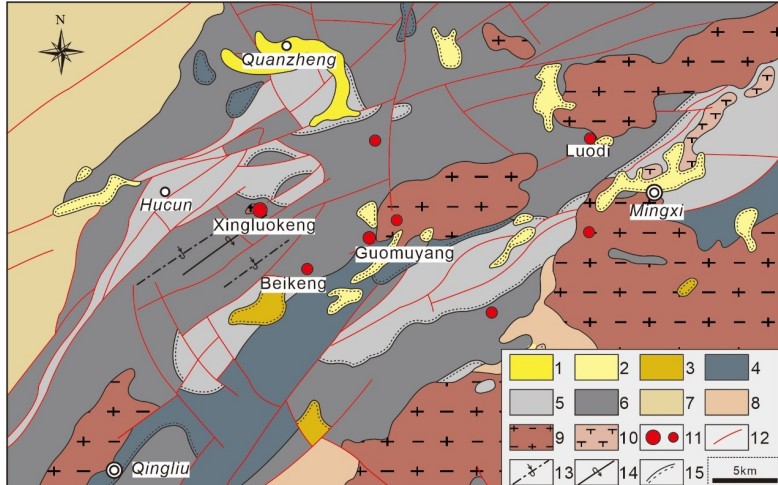

**Figure 2.** Regional geological map of the Xingluokeng W-Mo ore field showing W deposits in the area (modified after [46]). 1 = quaternary; 2 = tertiary; 3 = cretaceous; 4 = Jurassic System; 5 = Upper Devonian–Permian; 6 = Upper Sinian-Lower Cambrian; 7 = Caledonian gneissic biotite monzogranite; 8 = Indosinian gneissic biotite monzogranite; 9 = Yanshanian biotite granite; 10 = Yanshanian syenite; 11 = W deposit and mineralization point; 12 = fault; 13 = axis of overturned synclinal; 14 = axis of overturned anticline; and 15 = unconformity boundary.

## 3. Deposit Geology

The Xingluokeng W deposit is located about 35 km northeast of Ninghua County, western Fujian Province, and is the largest tungsten deposit in Fujian Province, featured by large reserves (~30 Mt of WO$_3$) and low grade (averaging at 0.23%). In addition to W, the reserves of Mo are 3.02 Mt with an average grade of 0.024%. With the deepening of the mining depth (the current mining depth is about 600 m elevation), Cu has also reached the industrial grade, the available exploration data show that there is still tungsten mineralization at 100 m elevation, and the Xingluokeng pluton is characterized by whole rock mineralization [32].

The outcropped stratigraphic sequences in the area are mainly Sinian Sanxizhai Formation (Z$_2$s) and Middle Devonian Tianwadong Formation (D$_3$t). The Sanxizhai Formation (Z$_2$s) comprises three sections from the bottom up: metamorphic siltstone with lenticular carbonates (Z$_2$s$^1$), metamorphic feldspar quartz sandstone (Z$_2$s$^2$), and metamorphic fine sandstone and siltstone (Z$_2$s$^3$). The NEE-trending Xingluokeng inverted anticline (dips to SE with an inclination of 43°–63°) controls the distribution of main stratigraphic units, with the first section (Z$_2$s$^1$) occurring at the core and the second section (Z$_2$s$^2$) at the wings. NEE-trending faults, as the dominant regional structures, control the emplacement of the Xingluokeng stock and are the main ore-hosting structures for large quartz vein-type ore bodies. A series of NW-trending faults was also developed, cross-cutting the Xingluokeng stock and the NEE-trending faults in addition to a number of near SN-trending faults (Figure 3). The Xingluokeng stock is composed of porphyritic biotite granite (G1) and medium- to fine-grained biotite granite (G2), which intruded into the north wing of overturned anticline along the interface between the first section and the second section of the Sanxizhai Formation [33]. The zircon U-Pb ages indicated they were emplaced at 152.5 ± 1.4 Ma and 152.2 ± 1.2 Ma and the geochemical data indicated that G2 is a moderately to highly fractionated I-type granite [30]. The late-stage granite porphyry and aplitic dikes are well developed in the mining area (Figure 3).

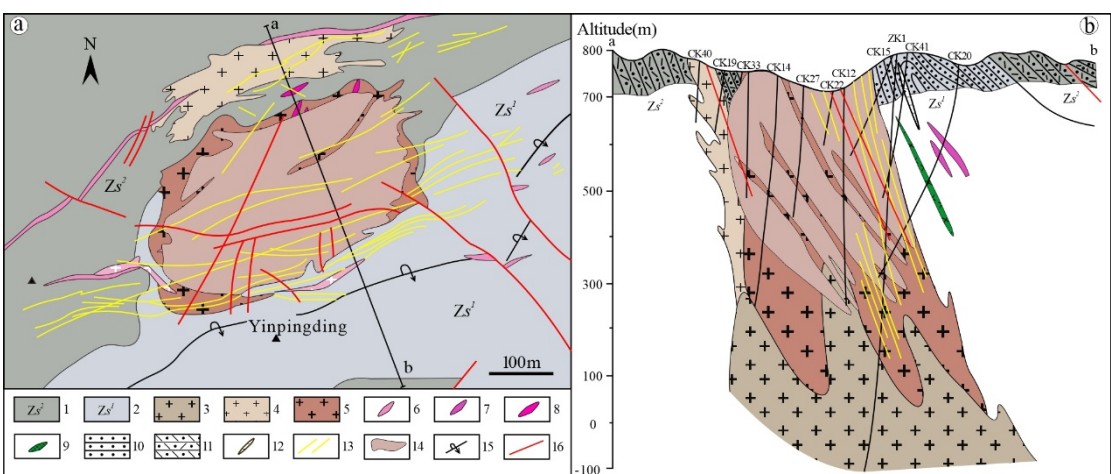

**Figure 3.** (**a**) Geological map of the Xingluokeng W-Mo polymetallic deposit (modified after [2]). (**b**) A NNW-SSE trending cross-section of the Xingluokeng W-Mo polymetallic deposit (modified after [33]). 1 = the second section of Sanxizhai Formation; 2 = the first section of Sanxizhai Formation; 3 = early Yanshanian fine-grained porphyritic granite; 4 = early Yanshanian fine- to medium-grained granite; 5 = early Yanshanian porphyritic biotite granite; 6 = granite-porphyry dyke; 7 = aplite dyke; 8 = borengite dyke; 9 = sillite dyke; 10 = hornfelsic metamorphic siltstone; 11 = hornfelsic tuff; 12 = dolomite limestone; 13 = mineralized quartz veins; 14 = axes of reverse anticline; and 15 = fault.

Two major mineralization styles have been identified at Xingluokeng: veinlet-disseminated and vein-type mineralization. The veinlet-disseminated mineralization mainly consists of disseminated molybdenite, scheelite, and wolframite, as well as minor pyrite and chalcopy-

rite in coexistence with quartz, beryl, fluorite, and muscovite, which are densely distributed in the altered granite and sparsely in the country rocks. The vein-type mineralization cross-cuts the veinlet-disseminated mineralization with ore minerals of wolframite, scheelite, and sulfides, in coexistence with quartz, K-feldspar, beryl, muscovite, fluorite, and calcite. Potassic alteration and greisenization are intensive and pervasive within the main orebody, overprinted by phyllic alteration, silicification, chloritization, and carbonatization assemblages. The Re-Os age of molybdenite was 156.3 ± 4.8 Ma [2] and the U-Pb age of wolframite were 151.3 ± 5.8 Ma and 150.5 ± 8.1 Ma [35].

The consistency of the chondrite-normalized REE distribution pattern of scheelite and mineralization-related granite in the Xingluokeng tungsten deposit indicated that the ore-forming fluid is mainly derived from the exsolution of magmatic fluid [33]. H-O and Sr isotopic compositions also suggested the ore-forming fluids dominantly originated from magma and limited meteoric water involved in the late mineralization stage [33].

## 4. Sampling and Analytical Methods

Wolframite and pyrite in quartz vein from the Xingluokeng W-Mo polymetallic deposit were collected for He-Ar isotope study. Samples were collected from different elevations in the open pit, and the detailed sampling information and sample descriptions are shown in Figure 4 and Table 1. The samples were crushed to 1~3 mm and the mineral (pyrite and wolframite) separates were handpicked under a binocular microscope.

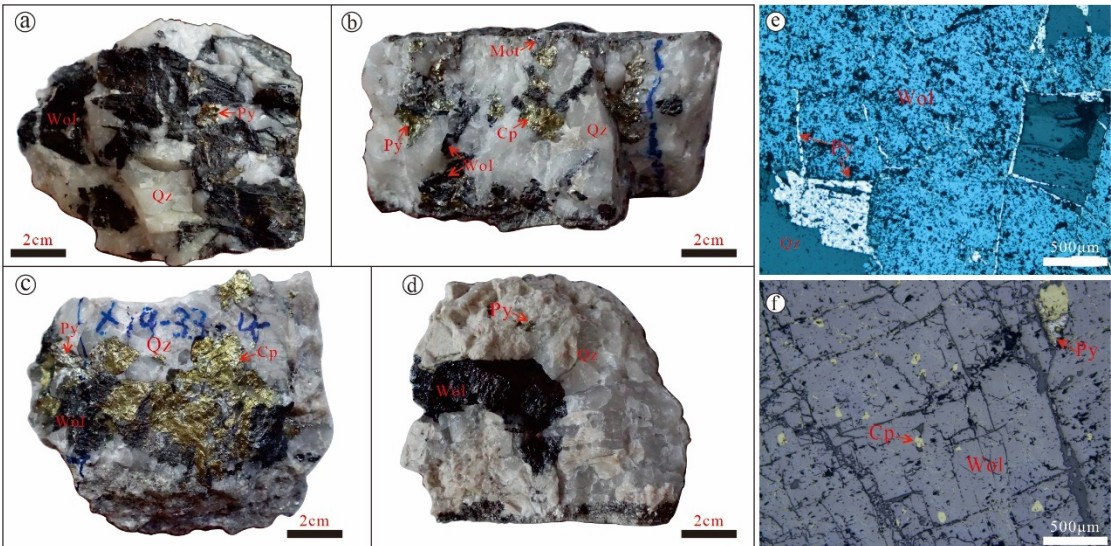

**Figure 4.** Photographs (**a**–**d**) and photomicrographs (**e**,**f**) of the Xingluokeng pyrite and wolframite samples. (**a**–**c**) are the pyrite samples used for He-Ar isotopic test, and the figures show that pyrite and other sulfides coexisted with wolframite; d is the wolframite sample used for He-Ar isotopic test, the figure show that wolframite was better crystallized; (**e**,**f**) are photomicrographs of pyrite and wolframite, which show that pyrite and wolframite were formed at the same metallogenic stage or slightly later than wolframite. Py = pyrite, Wol = wolframite, Cp = chalcopyrite, Mot = molybdenite, and Qz = quartz.

**Table 1.** Location and description of samples used for this study.

| Sample No. | Location | Mineral | Association |
|:---:|:---:|:---:|:---:|
| X19-10-1 | level 828 m | Pyrite | Pyrite associated with wolframite and quartz |
| X19-21-2 | level 744 m | Pyrite | Pyrite associated with wolframite, quartz, and muscovite |
| X19-28-2 | level 672 m | Pyrite | Pyrite associated with wolframite, molybdenite, and quartz |
| X19-32-1 | level 690 m | Pyrite | Pyrite associated with wolframite, quartz, and muscovite |
| X19-33-4 | level 690 m | Pyrite | Pyrite associated with wolframite, chalcopyrite, and quartz |
| X19-24-1 | level 720 m | Pyrite | Pyrite associated with quartz and chalcopyrite |
| X21-1-1 | level 672 m | Wolframite | Wolframite associated with quartz and feldspar |
| X21-1-2 | level 792 m | Wolframite | Wolframite associated with quartz and feldspar |
| X19-26-4 | level 708 m | Wolframite | Wolframite associated with quartz |

The He-Ar isotope analysis of fluid inclusions in minerals was carried out at the Beijing Institute of Geology of Nuclear Industry. The analytical procedures were similar to those described by [54,55]. First, the sample was cleaned in an acetone medium in ultrasonic waves for 20 min, then heated to 180 °C and degassed under vacuum for 48 h to remove the gas attached to the mineral surface, and the system vacuum was better than $10^{-8}$ Pa. Then, the sample was crushed to release the gas, and the released gas was purified in multiple stages by four zirconium aluminum pumps and an activated carbon cold trap to remove reactive gases such as $N_2$, $O_2$, $H_2$, and $CO_2$, and adsorb Ar, Xe, and Kr. The purified He fed into the system and was purified again by a titanium sublimation pump with liquid nitrogen, Ar was released at $-78$ °C, and He and Ar were fed into the mass spectrometer. The test instrument was Helix SFT rare gas isotope mass spectrometer produced by Thermo Fisher Scientific company of the United States, and the analytical error of the He-Ar isotope was less than 10%. Procedural blank contributions ($^4$He $< 1.12 \times 10^{-11}$ cm$^3$ STP, $^{40}$Ar $< 2.24 \times 10^{-11}$ cm$^3$ STP) were insignificant.

## 5. Results

The results of He-Ar isotopes of wolframite and pyrite are shown in Table 2. $^3$He/$^4$He values of fluid inclusions in 6 pyrite samples ranged from 0.14 Ra to 1.02 Ra (averaging at 0.76 Ra). The $^{40}$Ar/$^{36}$Ar values for these samples were 361.3 to 683.2, with an average of 544.8, the $^{40}$Ar values were 27.4 to 53.7 $\times 10^{-8}$ cm$^3$ STP g$^{-1}$ (averaging at 39.9 $\times 10^{-8}$ cm$^3$ STP g$^{-1}$), and the $^4$He values were 156 to 390 $\times 10^{-8}$ cm$^3$ STP g$^{-1}$ (averaging at 253.5 $\times 10^{-8}$ cm$^3$ STP g$^{-1}$). Compared with pyrite, the $^3$He/$^4$He values of 3 wolframite samples were relatively lower, ranging from 0.21 to 0.25 Ra. The $^{40}$Ar/$^{36}$Ar values for these samples were 539.6 to 816.7, with an average of 643.5, the $^{40}$Ar content was 15.0 to 58.0 $\times 10^{-8}$ cm$^3$ STP g$^{-1}$ (averaging at 37.9 $\times 10^{-8}$ cm$^3$ STP g$^{-1}$), and the $^4$He content was 292 to 646 $\times 10^{-8}$ cm$^3$ STP g$^{-1}$ (averaging at 498.7). The crushing method can extract the

noble gas in fluid inclusions of hydrothermal minerals, but one disadvantage of this analysis technology is that it is difficult to separate the fluid inclusions of different stages in minerals, and as a result, the experimental results represent the average value of inclusion-trapped fluids.

**Table 2.** He-Ar isotopic compositions of fluid inclusions trapped in pyrite and wolframite from the Xingluokeng W-Mo polymetallic deposit.

| Sample No. | X19-10-1 | X19-21-2 | X19-28-2 | X19-32-1 | X19-33-4 | X19-24-1 | X21-1-1 | X21-1-2 | X19-26-4 |
|---|---|---|---|---|---|---|---|---|---|
| Mineral | Pyrite | Pyrite | Pyrite | Pyrite | Pyrite | Pyrite | Wolframite | Wolframite | Wolframite |
| $^3$He ($10^{-14}$ cm$^3$STP/g) | 144 | 270 | 551 | 43 | 397 | 303 | 164 | 226 | 94 |
| $^4$He ($10^{-8}$ cm$^3$STP/g) | 156 | 212 | 390 | 218 | 278 | 267 | 558 | 646 | 292 |
| $^3$He/$^4$He ($10^{-7}$) | 9.24 | 12.74 | 14.14 | 1.96 | 14.28 | 11.34 | 2.94 | 3.50 | 3.22 |
| R/Ra ($\pm 1\sigma$) | 0.66 ± 0.01 | 0.91 ± 0.01 | 1.01 ± 0.01 | 0.14 ± 0.01 | 1.02 ± 0.01 | 0.81 ± 0.01 | 0.21 ± 0.01 | 0.25 ± 0.01 | 0.23 ± 0.01 |
| Mantle He (%) | 9.88 | 13.73 | 15.28 | 1.85 | 15.43 | 12.19 | 2.93 | 3.55 | 3.24 |
| $^{40}$Ar ($10^{-8}$ cm$^3$STP/g) | 53.7 | 27.4 | 46.1 | 31.6 | 47.9 | 32.7 | 58.0 | 40.8 | 15.0 |
| $^{40}$Ar/$^{36}$Ar ($\pm 1\sigma$) | 491 ± 0.4 | 683 ± 0.3 | 626 ± 0.4 | 361 ± 0.2 | 523 ± 0.3 | 582 ± 0.9 | 574 ± 0.8 | 816 ± 1.0 | 539 ± 0.7 |
| $^{38}$Ar/$^{36}$Ar ($\pm 1\sigma$) | 0.191 ± 0.003 | 0.19 ± 0.002 | 0.187 ± 0.003 | 0.191 ± 0.002 | 0.189 ± 0.003 | 0.193 ± 0.003 | 0.193 ± 0.002 | 0.188 ± 0.002 | 0.189 ± 0.002 |
| $^{40}$Ar* ($10^{-7}$) | 2.1 | 1.6 | 2.4 | 0.6 | 2.1 | 1.6 | 2.8 | 2.6 | 0.7 |
| $^{40}$Ar* (%) | 39.8 | 56.7 | 52.8 | 18.2 | 43.6 | 49.3 | 48.5 | 63.8 | 45.2 |
| $^3$He/$^{36}$Ar ($10^{-3}$) | 1.32 | 6.73 | 7.50 | 0.49 | 4.34 | 5.39 | 1.62 | 4.53 | 3.38 |
| $F^4$He | 8648 | 32,036 | 32,132 | 15,106 | 18,427 | 28,815 | 33,474 | 78,370 | 63,661 |

Mantle He (%) = [($^3$He/$^4$He)$_{sample}$ − ($^3$He/$^4$He)$_{crust}$]/[($^3$He/$^4$He)$_{mantle}$ − ($^3$He/$^4$He)$_{crust}$] × 100 [56], where ($^3$He/$^4$He)$_{crust}$ = 0.02 Ra, ($^3$He/$^4$He)$_{mantle}$ = 6.5 Ra [57]; $F^4$He = ($^4$He/$^{36}$Ar)$_{sample}$/($^4$He/$^{36}$Ar)$_{air}$, where ($^4$He/$^{36}$Ar)$_{air}$ = 0.165 [20]; and $^{40}$Ar* = $^{40}$Ar × [1 − ($^{40}$Ar/$^{36}$Ar)$_{air}$/($^{40}$Ar/$^{36}$Ar)$_{sample}$], where ($^{40}$Ar/$^{36}$Ar)$_{air}$ ≈ 295.5 $^{40}$Ar* (%) = $^{40}$Ar*/$^{40}$Ar.

## 6. Discussion

### 6.1. The Effect of Post-Ore Processes on He-Ar Isotopes

Helium and argon isotope compositions of fluid inclusions hosted by metallic minerals in ore deposits can be used to trace the contribution of mantle-derived fluids [24–26,28,58,59]. However, the post-metallogenic geological processes may affect the He and Ar isotope composition in the fluid inclusions and change the original information of ore-forming fluids; therefore, it is necessary to evaluate the degree of influence of post-metallogenic geological processes [60,61]. He isotopic composition can be affected by cosmogenic $^3$He, nucleogenic production of $^3$He, He loss and diffusiveness, etc. [18]. The $^3$He content of ore-forming fluids in minerals can be affected by cosmogenic $^3$He, but the influence range of cosmic rays is only within 1.5 m of the Earth's surface [19,58]. The samples collected in this study were all newly mined in the open pit, and thus the effect of cosmogenic $^3$He could be ignored. Nucleogenic $^3$He can be produced by the $^6$Li(n,α)$^3$H(β-)$^3$He reaction; however, due to the low concentration of Li in the fluid inclusions, the in situ production of $^3$He is negligible [21,23,62]. The radiogenic $^4$He in the fluid inclusion also negligible due to the low concentrations of U and Th [18,20,21,56]. Moreover, due to the influence of the stopping distance of α particles, radiogenic $^4$He has no influence on the $^4$He of fluid inclusions with diameters less than 20 μm [57]. He and Ar are released by crushing, which greatly reduces the influence of noble gases produced by elemental decay in the lattice [23].

The impact of $^{40}$Ar produced by the decay of $^{40}$K in the mineral lattice should be negligible because of the low diffusivity of Ar in pyrite [63,64] and the extremely low concentration of K in pyrite [64,65]. The $^{40}$Ar produced by the decay of K in fluid inclusions and the host mineral could be estimated by the equation $^{40}$Ar atoms g$^{-1}$ yr$^{-1}$ = 102.2 [K] [66], and the content of K in most tungsten deposits is about 1 ppm [67–69], and as a result, the radiogenic $^{40}$Ar is 5.7 × $10^{-10}$ cm$^3$ STP/g, which is several orders of magnitude lower than the total $^{40}$Ar content obtained in this study (average in 3.9 × $10^{-7}$ cm$^3$ STP/g), indicating that the effect of the decay of K in the fluid inclusions could be

neglected. In general, the measured $^{40}Ar/^{36}Ar$ ratios are lower than the true $^{40}Ar/^{36}Ar$ ratios of the fluids due to the contributions of atmospheric Ar [19]. Previous studies have confirmed that wolframite and pyrite are suitable minerals for the study of He-Ar isotopes [18–20,22,23,65]; therefore, the He and Ar isotopic values obtained in this study can represent the characteristics of ore-forming fluids.

### 6.2. Source of He and Ar

The sources of noble gases in hydrothermal minerals mainly includes ASW, mantle fluids, and crustal fluids, which have significantly different He-Ar isotopic compositions [19,66]. Compared with mantle fluids and crustal fluids, the amount of He in the ASW is extremely low, which is not enough to have a significant effect on the abundance and isotopic composition of He in fluids [23], supported by the high $F^4He$ values (6366~78370, Table 2); therefore, He in ore-forming fluids mainly comes from the mantle or crust. The $^3He/^4He$ ratios of mantle fluids are significantly higher than those of crustal fluids, where the $^3He/^4He$ ratios of the upper mantle is 7–9 Ra [70], the subcontinental lithospheric mantle (SCLM) is 6–7 Ra [71], and the crust is 0.01~0.05 Ra [72]. The $^3He/^4He$ ratios of pyrite and wolframite in the Xingluokeng W-Mo polymetallic deposit were 0.14~1.02 Ra and 0.21~0.25 Ra, respectively, which are higher than the crustal fluid ratio but lower than the mantle fluid ratio, indicating that the He in the ore-forming fluids was a mixture of crustal fluids and mantle components. According to the estimation formula [66], the proportions of mantle-derived He in pyrite and wolframite were 1.85~15.28% and 2.93~3.24%, respectively, indicating that the He in the ore-forming fluids was mainly derived from the crust and minorly from the mantle.

The $^{40}Ar/^{36}Ar$ ratios of pyrite and wolframite in the Xingluokeng deposit were 361.3~683.2 and 539.6~816.7, respectively, which are higher than the value of air-saturated water (ASW, $^{40}Ar/^{36}Ar$ = 295.5) and indicate the presence of radiogenic $^{40}Ar$ derived from mantle or crustal components. The radiogenic $^{40}Ar^*$ ($^{40}Ar^* = ^{40}Ar - [^{36}Ar \times 295.5]$) could be estimated using the equation provided by [73]. The estimated $^{40}Ar^*$ of pyrite and wolframite were 18.2~56.7% and 45.2~63.8%, respectively, and correspondingly, the $^{40}Ar$ from the air is 43.3%~81.8% and 36.2~54.8%, respectively.

The linear correlations between $^3He/^{36}Ar$ and $^{40}Ar/^{36}Ar$ (Figure 5), and between $^3He/^4He$ and $^{40}Ar^*/^4He$ (Figure 6) indicated that the mineral-hosted volatiles from the Xingluokeng deposit were mixtures of two fluids: a high $^3He/^4He$-$^{40}Ar/^{36}Ar$-containing mantle component and a low $^3He/^4He$-$^{40}Ar/^{36}Ar$ crustal component. Extrapolating the trend in Figure 5 to pure ASW ($^3He/^{36}Ar = 5 \times 10^{-8}$) generated a $^{40}Ar/^{36}Ar$ value of 446, which was a little higher than that of ASW (295.5), indicating that the crustal component had near atmospheric $^{40}Ar/^{36}Ar$. Therefore, the crustal fluid was probably modified air-saturated water (MASW) with a crustal $^3He/^4He$ ratio and near atmospheric $^{40}Ar/^{36}Ar$. The other was a fluid exsolved from the W-associated granitic magma. This is consistent with the conclusion of REE characteristics and H-O and Sr isotopes that the ore-forming fluids dominantly originated from magma water and meteoric water involved in the late mineralization stage [33].

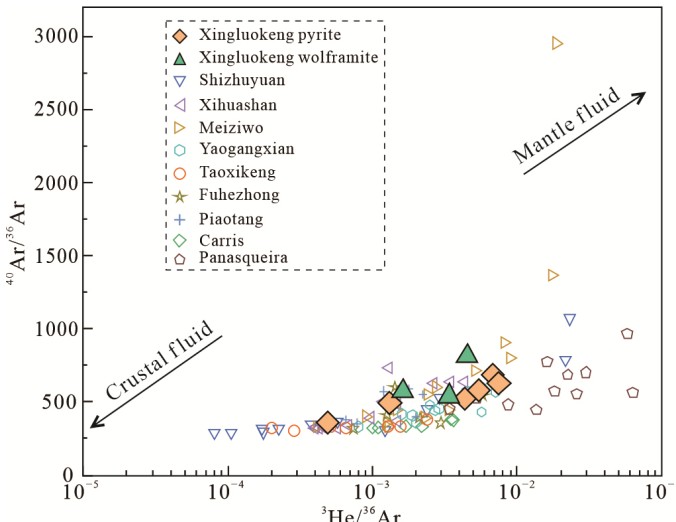

**Figure 5.** $^{40}Ar/^{36}Ar$ vs. $^3He/^{36}Ar$ plot of the ore-forming fluids from the Xingluokeng W-Mo polymetallic deposit. The figure also shows the published He-Ar isotopic compositions of a number of other W deposits around the world, such as Shizhuyuan [25], Xihuashan [24], Meiziwo [28], Yaogangxian [18], Taoxikeng [27], Fuhezhong, and Piaotang [26] W deposits in south China, and the Carris W-Mo-Sn deposit [22] and Panasqueira in Portugal [20]. The values of mantle fluids and crustal fluids are quoted from [18].

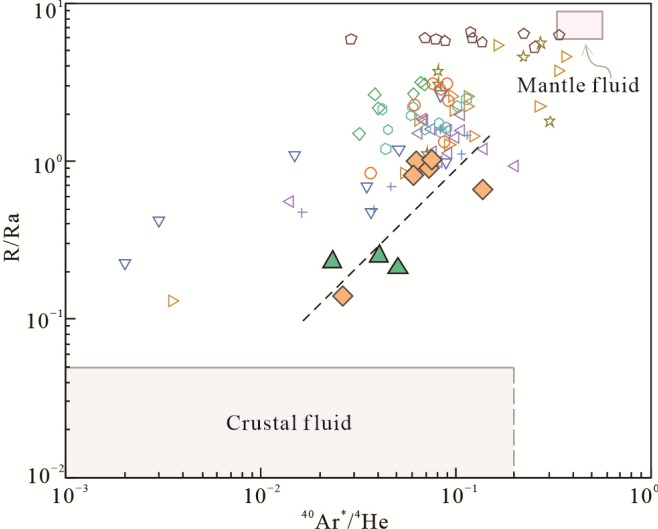

**Figure 6.** R/Ra vs. $^{40}Ar^*/^4He$ plot of the ore-forming fluids from the Xingluokeng W-Mo polymetallic deposit. The values of mantle fluids and crustal fluids are quoted from [18]. The linear correlation was obtained by least squares regression: $^3He/^4He$ (Ra) = $6.0 \times ^{40}Ar^*/^4He + 0.22$, $r^2 = 0.69$. Symbols are the same as for Figure 5.

### 6.3. Role of Mantle Upwelling in Granite-Related W Mineralization in South China

The Xingluokeng W-Mo deposit had similar $^3He/^4He$ ratios with large or super-large W deposits in the NLMB, such as the Xihuashan W deposit and Shizhuyuan W-Sn-Bi-Mo deposit, indicating similar origin of noble gases in the ore-forming fluids of the W deposit in NLMB and WYMB. However, they are far lower than that of the Zijinshan high-sulfidation Cu-Au deposit [17] in the WYMB, which with significant mantle contributions, indicate different geneses of W deposits and Cu-Au deposits in the WYMB. Most W deposits have $^3He/^4He$ ratios lower than 2 Ra (Figure 6), indicating a primary contribution of the crust-derived fluids in the genesis of ore deposits [24–26,74]. However, the He-Ar isotopic

compositions showed that evident mantle contribution occurred during the metallogeny of W deposits both in south China and the world (Figures 5 and 6). Especially, the high $^3$He/$^4$He ratios (up to 4.53 Ra) of W deposits in the Yaoling-Meiziwo and Fuhezhong area in south China and the high $^3$He/$^4$He ratios (up to 6.7 Ra) in the Panasqueira W deposit indicate a significant contribution of mantle-derived components [20,28,75].

Noble gases in the mantle are generally trapped in minerals. If there is no generation and transportation of mantle magma, rare gases and volatiles in the mantle can hardly reach the surface through diffusion, because the diffusion distance of volatiles is limited even at the mantle temperatures [66]. Therefore, the occurrence of mantle-derived $^3$He in the ore-forming fluids is often the reflection of the generation and the degassing of mantle magma [76,77]. The mantle-derived components involved in the ore-forming fluid indicate that intrusion of mantle-derived magma occurred during the formation of large-scale tungsten mineralization and related Yanshanian granite in south China, which either directly entered the magma during the remelting of the crust or was injected into the magma chamber at the late stage of magma formation. However, the ore-related magma properties are still dominated by crustal materials.

The dynamics of the Mesozoic diagenesis and mineralization in south China is still debated, but it is generally accepted that the large-scale Mesozoic diagenesis and mineralization in south China is closely related to the lithospheric extension [42,78–80]. The metallogeny of tungsten and related granites in south China are concentrated in 160–150 Ma [81,82]. It is suggested that from 160 to 150 Ma, south China underwent lithospheric extension-thinning coupled with mantle upwelling [43,83–86], which is closely related with the tear-off or rollback of subducted Paleo-Pacific plate [82,87]. The deep dynamics of mantle upwelling and the reasons for the difference in the intensity of crust–mantle interactions in different areas need further study, but the upwelling of asthenospheric mantle and crust–mantle interactions between 160 Ma and 150 Ma are generally recognized. The upwelling of mantle magma leads to the melting of the W-Sn-rich ancient crust to form the initial magma (Figure 7), which is further enriched in the ore-forming fluids during the process of magma crystallization and differentiation. The W-Sn-rich ancient crust provides the mineralization material, and the upwelling mantle magma provides heat and various proportions of materials or volatiles for the ore-forming magma and hydrothermal system. The heat continuously provided from the mantle can reduce the rate of magmatic condensation and facilitate sufficient differentiation of the magma so that ore-forming elements are fully enriched into the ore-forming fluid. Most of the W deposits in south China span several Ma [7], much longer than individual intrusion-sustained hydrothermal activity [20], and this may require continuous heating from the mantle.

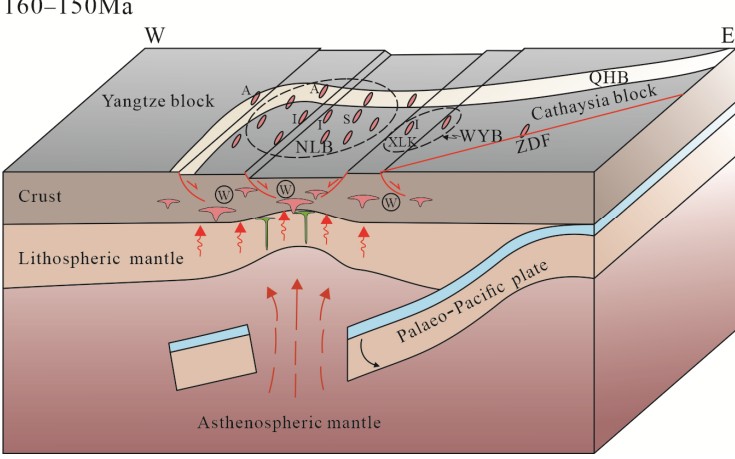

**Figure 7.** Cartoon showing the geodynamic setting of W mineralization in south China (modified after [78]).

### 7. Conclusions

The $^3$He/$^4$He ratios of the minerals form Xingluokeng deposit ranged from 0.14 to 1.01 Ra with an average of 0.58 Ra, and the $^{40}$Ar/$^{36}$Ar ratio ranged from 361 to 817 with an average of 578. The mantle-derived He was found to be added to the ore-forming fluid, with an average proportion of 8.7%, up to 15.43%. The ore-forming fluid of the Xingluokeng deposit was a mixture of the crustal fluid, which was probably modified air-saturated water (MASW) with a crustal $^3$He/$^4$He ratio and near atmospheric $^{40}$Ar/$^{36}$Ar, and magmatic fluid exsolved from the W-associated granitic magma. The Xingluokeng W-Mo polymetallic deposit as well as other W deposits in NLMB and WYMB were formed under the Mesozoic extensional tectonic background, where the upwelling mantle provided heat, volatiles, and probably various degrees of materials to the W-related granites. The upwelling mantle plays an important role in the formation of W deposits.

**Author Contributions:** Writing—original draft preparation, Y.G.; sample collection, G.Z. and J.S.; methodology, Y.G.; article writing and figure drawing, B.C. and L.W.; writing—reviewing and editing, J.G. All authors have read and agreed to the published version of the manuscript.

**Funding:** This work was jointly supported by the National Key R&D Program of China (2017YFC0602602 and 2016YFC0600207).

**Data Availability Statement:** Not applicable.

**Acknowledgments:** We express our sincere thanks to the Xingluokeng Tungsten Mine Co., Ltd. for their valuable support during the field geological investigation.

**Conflicts of Interest:** The authors declare no conflict interest.

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
