# Peer review of "Mantle-Derived Noble Gas Isotopes in the Ore-Forming Fluid of Xingluokeng W-Mo Deposit, Fujian Province"

_minerals, doi:10.3390/min12050595_

Round 1
Reviewer 1 Report
Dear Authors,
Please find enclosed my comments on the manuscript.
Mantle-derived noble gas isotopes in the ore-forming fluid of Xingluokeng W-Mo deposit, Fujian Province,
by Gao et al.
This paper reports isotopic compositions of He and Ar extracted from ore minerals in the Mesozoic W-Mo deposit in China. The authors discussed involvement of mantle-derived noble gases in the ore-forming fluid, which might be a clue to explore the origin of the deposits. The analysis data might have potential to be disclosed, but the discussion about quantitative evaluation of the effects of secondarily generated nuclides especially on 3He/4He is insufficient. Though the authors have devoted lots of pages to discuss the possible occurrence of mantle-derived components, they do not succeed in total elimination of the influence of the post-eruptive accumulation of radiogenic/nucleogenic nuclides. The authors should clear up the anxiety. Consequently I suggest major revision especially of the discussion about the influence of secondarily generated nuclides.
Major comments:
1. Post-eruptive accumulation of radiogenic nuclides.
The authors argue that the contribution of radiogenic 4He and 40Ar is negligible. However, the samples in this study are quite old for the measurement of noble gas isotope ratios. The influence of secondary radionuclides should be carefully assessed even in the extraction of noble gases by the crushing method (Yokochi et al., 2005 GGG 6, doi:10.1029/2004GC000836).
2. The effect of nucleogenic nuclides.
3He can be generated within minerals based on the nuclear reactions in the Earth: 6Li(n,α)3H(β)3He. Therefore, it is essential to evaluate the effect if the authors would like to discuss the occurrence of mantle-related high 3He/4He. In this study, Figure 5 is essential as evidence of the contribution of the mantle. 40Ar and 3He correlations may be due to the mantle contribution. However, their parent nuclides, 40K and 6Li, are both alkaline elements, both are similar in behavior to each other. This means that the correlation in Figure 5 can be explained by the addition of radiogenic 40Ar and nucleogenic 3He to an ore-forming fluid without a mantle component. If this possibility can be ruled out, it would be safe to conclude that the mantle contributes to this ore-forming fluid.
There are some other places in which some clarification is needed as described below.
lines 185-186. The uncertainty in isotope ratios is an important information for this study. The uncertainty in the isotope ratio varies greatly depending on the amount of gas used in the measurement. They should be properly listed in the table.
lines 222-224. No evidence of low concentrations of U, Th, or K in the samples used in this study was provided.
lines 233-234. At this point, it can also be explained by the addition of radiogenic 4He to the ASW.
Table 2. It is necessary to add errors in isotope ratios. In addition, the description of 3He/4He is flawed.
Figure 5 and 6. No citation is given for the Mantle fluid and Crustal fluid values.
Best regards,
Reviewer 2 Report
The manuscript reports He and Ar isotopic compositions of fluid inclusions in pyrite and wolframite from the Xingluokeng W-Mo deposit, aiming to evaluate the origin of ore fluids and discuss the contribution of the mantle to tungsten mineralization. The data and figures are pretty good, and the manuscript is well organized and written. The manuscript can be accepted after minor revision. My comments are attached and I wish they are helpful for the revision.
Major comments:
One target of this article is to evaluate the contribution of mantle components to W-Mo mineralization by He-Ar isotopes. However, most casual igneous rocks for W-Mo mineralization are felsic with a crustal source. This study also shows that the calculated ratio of mantle-derived He in the ore fluid is very low (8.7-15.43%). Given this condition, the mantle contribution to ore fluid and metal is not important, and the relations between mantel component and mineralization should be indirect. If this is true, the pivotal scientific problem of this article should be adjusted moderately.
Specific comments:
Line 13, china should be China?
Line 22, 8.7% and up to 15.43%, different Significant Figures, Why?
Figure 1, Coordinate for Figure 1a is not right. Nanling metallogenic belt? Nanling Range? The text and figures should be consistent. Zijinshan should be labeled in Figure 1b.
L72, the Jiangnan Orogen is different from the Qin-Hang metallogenic belt in figure 1b.
L83, Jura?
L88, WYMB is different from the WYB in figure 1b.
Figure 3, the ages of intrusions can be illustrated. In addition, where is the Middle Devonian Tianwadong Formation?
Table 1, Which mineralization stage/ mineralization styles are these samples from?
18/04/2022
Round 2
Reviewer 1 Report
Dear Mr. Kyle Liu,
Please find enclosed my comments on the manuscript.
Mantle-derived noble gas isotopes in the ore-forming fluid of Xingluokeng W-Mo deposit, Fujian Province,
by Gao et al.
This paper reports isotopic compositions of He and Ar extracted from ore minerals in the Mesozoic W-Mo deposit in China. It is second review for the manuscript. I found the authors to be effective in responding to my comments raised at the first review. Although the influence of the secondary generation of3He and4 He has not been completely eliminated, the number of references provided helped to minimize the concern. Consequently this paper is suitable for publication in Minerals.
Best regards,
Reviewer